# Penile Rehabilitation after Prostate Cancer Treatment: Which Is the Right Program?

**Roberto Castellucci** [1,*] **, Piergustavo De Francesco** [1] **, Antonio De Palma** [2] **, Davide Ciavarella** [2] **, Simone Ferretti** [2] **, Michele Marchioni** [1,2] **and Luigi Schips** [1,2]

1. Department of Urology, ASL Abruzzo 2, 66100 Chieti, Italy
2. Unit of Urology, Department of Medical Oral and Biotechnological Sciences, SS. Annunziata Hospital, G. d'Annunzio University, 66100 Chieti, Italy
* Correspondence: roberto.castellucci@gmail.com; Tel.: +39-3295449222; Fax: +39-0871357756

**Abstract:** The management of sexual complications after treatment of localized prostate cancer, such as erectile dysfunction, changes in the length of the penis, pain during sexual intercourse, and lack of orgasm, is still an unsolved problem with an important impact on patients' quality of life. In this review, we summarize the current scientific literature about the rehabilitation of erectile dysfunction after prostate cancer treatment. The therapy for penile rehabilitation includes different types of treatments: the combination of phosphodiesterase type 5 inhibitors (PDE5-I) and the vacuum erectile device (VED) are considered first-line treatment options. When therapy begins, the duration of treatment, the dosage and the drug used all play very important roles in the treatment outcome. Intracavernous injection (ICI) therapy represents the second-line option for patients ineligible for PDE5-I therapy. Technological development has led to the emergence of devices for the stimulation of the penis without the use of drugs, such as penile vibratory stimulation (PVS) for stimulation of ejaculation in spinal cord injury and low-intensity extracorporeal shockwave therapy (LIESWT). The rapid diffusion of the latter, thanks to its easy use, attains good results without side effects. The panorama of penile rehabilitation after PC treatments is vast and many studies are needed, especially on new technologies, to find the best therapeutic regimen possible, personalized to the patient's characteristics and the type of treatment for PC.

**Keywords:** prostate cancer; prostate cancer treatment; radiotherapy; castration therapy; radical prostatectomy; erectile disfunction; low-intensive shockwave therapy; penile rehabilitation





## 1. Introduction

Prostate cancer (PC) is the second most commonly diagnosed cancer in men with 1.4 million new diagnoses in 2020 in the world [1]. Treatment of localized PC can be done in various ways: active surveillance (AS), radical prostatectomy (RP), radiation therapy (RT), androgen deprivation therapy (ADT), and multimodal treatments. Newly available technologies and greater surgical experience have led to a higher survival rate in men with localized PC [2]. Of the problems that may arise from the treatment of prostate cancer, urinary incontinence and erectile dysfunction (ED) are certainly the most important [3]. These alterations have an important impact on the patient's quality of life. Despite the newly available technologies for the treatment of PC, the secondary effects of urinary incontinence, such as urinary urgency, an overactive bladder, detrusor overactivity, and, within the sexual sphere, erectile dysfunction, changes in the length of the penis, pain during sexual intercourse, lack of orgasm, lack of self-confidence, and a decrease in libido, remain important problems for doctors and patients [4]. For the treatment of urinary incontinence, male slings and artificial urinary sphincters were the most common procedures [5]. Reports from high-volume centers showed that 3 months after RP the proportion of potent patients ranges from 8% to 65% based on the level of

nerve-sparing surgery [6]. Patients who undergo external beam radiation therapy may develop more severe ED than those who undergo brachytherapy [7]. Iacono et al. evaluated the histological changes in the corpora cavernosa before and after radical prostatectomy and demonstrated that the amount of collagen tissue increased in the postoperative period [8]. For that reason, it is essential to undertake penile rehabilitation. Penile rehabilitation is a medical intervention that aims to preserve the integrity of the penile structures after PC therapy. The main treatments used for penile rehabilitation are type 5 inhibitors of phosphodiesterase (PDE5-I), intracavernous injection (ICI) of vasa-active agents and vacuum erectile device (VED), low-intensity extracorporeal shockwave therapy (LIESWT), and stem cell therapy (SCT) [9]. Other new treatments could be found in the future of gene-based therapy and platelet-rich plasma. The purpose of this review is to summarize the scientific evidence relating to the penile rehabilitation of patients undergoing prostate cancer therapy.

## 2. Materials and Methods

This article is based on a review of the PubMed scientific literature for everything related to the rehabilitation of ED after PC treatment up to April 2022.

The keyword search included prostate cancer, prostate cancer treatment, radiotherapy, castration therapy, radical prostatectomy, erectile dysfunction, low-intensive shockwave therapy, and penile rehabilitation. Presentations at courses and conferences or video communications were excluded.

### 2.1. PDE5 Inhibitors (PDE5-I)

Oral phosphodiesterase type 5 inhibitors (PDE5-I) have revolutionized the treatment of ED since 1998. With a very good safety profile, PDE5-I inhibits a very important enzyme (PDE5), which is involved in the conversion of guanosine monophosphate (GMP). This mechanism results in an increased level of cyclic guanosine monophosphate (cGMP) in the penis smooth muscle which leads to a more powerful and sustained erection [10]. A lot of studies have investigated the role of PDE5-I after RP or RT in erectile function (EF) after one of these primary treatments, as shown in Table 1.

**Table 1.** Studies that have investigated the role of PDEI-5 after RP or RT in EF after primary treatments.

| Studies | N. Patients and PC Treatment (PR or RT) | Type of Trial and Follow Up Months (m) | DE Treatment | Findings |
|---|---|---|---|---|
| Padma Nathan et al. (2008) [11] | 125 RP | Prospective, double blind, randomized 11 m | Daily Sildenafil 50/100 mg vs. placebo | Higher IIEF and night rigidity in sildenafil group |
| Montorsi et al. (2008) [12] | 628 RP | Prospective, double-blind, randomized 13.5 m | Daily Vardenafil 5/10 mg vs. on-demand Vardenafil 10/20 mg vs. placebo | Higher IIEF in on-demand group but after washout no differences |
| Pavlovich et al. (2013) [13] | 100 RP | Prospective, double-blind, randomized 13 m | Daily Sildenafil 50 mg with on-demand placebo vs. daily placebo with on-demand sildenafil | No differences between groups |
| Mulhall et al. (2013) [14] | 298 RP | Prospective, double-blind, randomized 3 m | Avanafil on-demand 100 mg vs. Avanafil 200 mg vs. placebo | Erectile functions improved in Avanafil groups 100 or 200 |

**Table 1.** *Cont.*

| Studies | N. Patients and PC Treatment (PR or RT) | Type of Trial and Follow Up Months (m) | DE Treatment | Findings |
|---|---|---|---|---|
| Montorsi et al. (2014) [15] | 423 RP | Prospective, double-blind, randomized 13.5 m | Daily Tadalafil 5 mg vs. on-demand tadalafil 20 mg vs. placebo | Higher IIEF in tadalafil daily but after washout non differences |
| Ilic et al. (2012) [16] | 27 RT | Prospective, double-blind, randomized 24 m | Daily Sildenafil 50/100 mg vs. placebo | No improved erectile functions in sildenafil group |
| Zelefsky et al. (2014) [17] | 279 RT | Prospective, double-blind, randomized 24 m | Daily Sildenafil 50 mg vs. placebo | Erectile functions improved in sildenafil group |
| Pisansky et al. (2014) [18] | 242 RT | Prospective, double-blind, randomized 12 m | Daily Tadalafil 5 mg vs. placebo | No improved erectile functions in tadalafil group |

The PDE5-I used most often were sildenafil, tadalafil, vardenafil, and avanafil. The first trial was conducted in 2008 by Padma et al.; it consisted of the administration of daily sildenafil at doses of 50 or 100 mg for 36 weeks, starting 4 weeks after RP in 125 patients. This study showed a higher International Index of Erectile Function (IIEF) in the sildenafil group and a higher incidence of nocturnal erections, but only 76 patients completed eh trial [11]. Montorsi et al. in the same year conducted a double-blind randomized trial involving 628 patients after bilateral nerve-sparing RP (NSRP) all around the world (REINVENT study). For these patients, vardenafil was administered daily (5 or 10 mg) or on-demand (10 or 20 mg) and compared to a placebo group. The on-demand group had a higher IIEF compared with the daily vardenafil and placebo group, but, after washout, there were no significant differences between the groups [12]. Similar results using sildenafil were found by Pavlovich et al. in 2013; 100 patients were randomized into a nightly sildenafil group (nightly sildenafil 50 mg vs. on-demand placebo) and an on-demand sildenafil group (daily placebo vs. on-demand sildenafil 50 mg) at the start of the day after RP for 12 months. No differences in the two groups were observed in EF in the follow-up period. This study had several limitations, including the short follow-up period, the lack of a pure placebo arm, and the recruitment of mainly Caucasian patients [13]. Mulhall et al., in their randomized trial, recruited 298 patients to whom avanafil was administered on-demand (100 mg and 200 mg) after RP for 3 months. After this period, the results showed that avanafil on-demand improved the EF of these patients significantly compared to the placebo. However, there was also a limit to this study; the long-term effects of this therapy were not assessed [14]. One of the most important trials in this field is the REACTT trial by Montorsi et al. After NSRP, 423 patients were randomized into either a tadalafil 5 mg daily group, a tadalafil 20 mg on-demand group, or a placebo group, followed by a washout period (6 weeks) and then followed by 3 months of tadalafil 5 mg daily in all patients. After 9 months, the EF was improved in the daily tadalafil group significantly, but after the washout period, there were no differences between groups. After the 3 months of tadalafil 5 mg daily for all patients there was an increase in erections in all treatment groups. Another very important endpoint in this trial was the consideration of penile length; in the daily tadalafil group, there was less loss in penile length compared to other groups. This fact suggests that PDE5-I could preserve penile integrity after post-prostatectomy neural injury [15,19]. Considering the post-prostatectomy field and in addition to these trials, many other studies have been published with important results. In 2015, Patel

conducted an analysis of REACTT patients, evaluating several aspects of their QoL using various questionnaires, such as the "Expanded Prostate Cancer Index Composite Short Form (EPIC-26)", the "Erectile Dysfunction Inventory of Treatment Satisfaction (EDITS)", and the "Self-Esteem and Relationship (SEAR)". All these questionnaires showed improved results in the tadalafil daily group but not in the tadalafil on-demand group [20]. Another important study was conducted by Jo et al. In this study, the authors randomized the start of PDE5I (sildenafil) soon after PR vs. 3 months after surgery. This trial showed that time is very important in penile rehabilitation; patients who received sildenafil soon after surgery achieved a significant increase in EF compared to those who received PDE5I three months after RP [21]. While the mechanism involved in ED post-RP seems to be more related to neural injury, the post-RT ED may be related to endothelial dysfunctions. One of the first trials in this field was published by Ilic et al. The authors administrated sildenafil (50 or 100 mg) or a placebo in 27 patients soon after RT. In the first week of treatment, there was an increase in EF in the sildenafil group, but this result was not confirmed at the end of the follow-up (24 months). This trial was very weak due to the small number of patients involved [16].

Zelefsky et al. recruited 279 RT patients with or without ADT; these patients were randomized and administered either sildenafil 50 mg daily or a placebo, starting 3 days before RT for 6 months. Obviously, ADT patients had worse EF but in no-ADT-patients, there was a significant improvement in EF in the sildenafil group after 24 months (81.6% of the sildenafil group and 56% of placebo patients reported functional erections) [17]. Another important study was published by Pisansky et al. This was the first multicentre, stratified, placebo-controlled, double-blinded, parallel-group study in this field; 242 patients were randomized to take tadalafil 5 mg daily vs. a placebo, seven days after RT for 6 months. In the follow-up period, there was no difference between the two groups and no improvement in EF. The limit of this study was the tadalafil dosing schedule; in addition, a longer follow-up and a larger cohort study could probably lead to a different result [18].

Based on the results of these studies, the question of the real effectiveness of PDE5-I after RP or RT is still open. Run Wang et al. in 2017 tried to answer this question with a meta-analysis, considering only the post-RP studies. This analysis showed that PDE5-I were very effective in the treatment of ED post-RP and preoperative mild ED, especially in higher dosages, longer durations of treatment, and in the use of sildenafil [22]. More recently, in 2021, Jia Wang et al. published a network meta-analysis, involving 14 randomized controlled trials, about the use of PDE5-I (all types) after RP. This study revealed that Avanafil 200 mg on-demand was the best therapy for ED after RP, but it recommended that more randomized controlled trials were needed to validate this consideration [23]. In 2022 Kang et al. published a double-blind prospective pilot study in which, for the first time, PDE5-I was administered before RP. Patients were randomized into two groups: tadalafil 5 mg once daily for 2 weeks before RP and tadalafil 5 mg once daily for 4 weeks after RP. The results showed that starting tadalafil before RP could be a safe and effective method to reach a better erection after RP, although the small study size is a limitation (only 41 patients were included) [24].

### 2.2. Intracavernous Injection (ICI)

Intracavernous Injection therapy is an important therapeutic option for men with erectile dysfunction, particularly after a radical prostatectomy. ICI stimulates the relaxation of the smooth muscles of the corpora cavernosa, generating an erection. The most administered drugs are phentolamine, prostaglandin E1 (PGE1), and papaverine. Phentolamine is an alpha-adrenergic antagonist that generates vasodilation. PGE1 is an endogenous prostaglandin that allows for the production of cAMP, which reduces the influx of calcium into the vascular smooth muscle of the penis, leading to its relaxation. Papaverine is a direct-acting smooth muscle relaxant, causing inhibition of the PDE enzyme and calcium channels [25].

ICI guarantees stable and long-lasting erections and does not require external devices, such as a vacuum erection device (VED), and has tolerable side effects. However, some studies report a high dropout rate (11–31%), mainly due to the possible ineffectiveness or excessive duration of the erection that can easily exceed 2 h and be painful. The need to inject drugs directly into the corpora cavernosa also does not make this treatment well tolerated in the long term [25]. Oral PDE5-I, possibly in combination with VED, is considered the first-line option for penile rehabilitation for patients who have undergone nerve-sparing RP; however, ICI therapy is an important second-line option [26]. Domes et al. found that ICI therapy was associated with significantly higher IIEF-5 scores in patients who failed oral PDE-5 inhibitor therapy post-RP. ICI therapy has also been studied in combination with PDE-5 inhibitors in patients with suboptimal responses to monotherapy [27], as shown in Table 2.

**Table 2.** Characteristics of the VED and ICI studies analyzed.

| Studies | Studies Type | Treatment | N° Patients | Surgery | Age (Years) | Follow-Up (Months) | Aim of Studies |
|---|---|---|---|---|---|---|---|
| Domes et al., 2012 [27] | Retrospective RCT | ICI | 117 | NSRP | $65 \pm 6.2$ | 12 | IIEF-5 score increase |
| Deng H et al., 2017 [28] | Retrospective RCT | No Therapy, PDE5, PDE5 + VED | 71 (34, 23, 14) | NSRP | 18–70 | 3 6 12 | IIEF-5 score increase |
| Raina et al., 2006 [29] | Retrospective RCT | VED | 109 | NSRP No NSRP | 58.2 | 9 | IIEF-5 score increase, penis length variation |
| Monga et al., 2006 [30] | Retrospective RCT | VED (A) 1 month later (B) 6 months later | 28 | NSRP | (A) 58.2 (B) 60.5 | 12 | IIEF-5 score increase, penis length variation |
| Mydlo et al., 2005 [31] | Retrospective RCT | ICI | 34 | NSRP | 46–66 | 12 | IIEF-5 score increase |

Mydlo et al. found that ICI therapy was useful in maintaining the efficacy of the PDE-5 inhibitor and 36% of patients were able to reduce ICI use after 7 months of treatment because the quality of their erections had improved [31]. Mosbah et al. also found a significant increase in overall satisfaction, sexual satisfaction, and IIEF-5 scores in patients who started penile rehabilitation 2 months after RP compared to patients who started penile rehabilitation 6 months after RP. [32].

### 2.3. Vacuum Erection Devices (VED)

VED is an important component of penile rehabilitation protocols. The physiological rehabilitative principle is that the artificial induction of erections immediately after surgery facilitates tissue oxygenation, reducing cavernous fibrosis and penis shortening in the absence of nocturnal erections, potentially increasing the preservation of erectile function [33]. It consists of a transparent plastic cylinder closed at one end and a vacuum pump capable of generating a negative pressure of 150–200 mmHg. It is positioned at the base of the penis and has the ability to generate artificial erections by sucking arterial and venous blood into the corpora cavernosa regardless of nerve disorders [33].

VED can be used both for rehabilitation alone and for having sexual intercourse. To have intercourse it is necessary to use a constriction ring positioned at the base of the penis that allows you to have a prolonged erection. However, this ring decreases the oxygen saturation of the tissues after 30 min of use. The use of the constriction ring is therefore recommended for intercourse but not for rehabilitation therapy [34]. Many recent clinical studies have indicated that early VED use after RP is a simple and effective method for penile rehabilitation. VED also increases the patient's confidence and enthusiasm and the sexual satisfaction of his partner, maintains penis length, and allows for an early return of spontaneous erection for post-RP men [35].

Liu's meta-analysis revealed that the use of VED was an important supportive therapy to drug therapy. Reduction of tissue fibrosis and loss of smooth muscle cells are the main goals of rehabilitation therapy. VED allows for erections earlier than PDE5-I alone, and

when vacuum therapy is combined with PDE5-I the benefits of penile rehabilitation appear to be significantly better [36,37]. Additionally, vacuum therapy can be used as an alternative treatment if PDE5-I are ineffective or contraindicated [28].

The most recent study on the use of VED is the Deng H et al. in 2017 [28]. It investigates the use of PDE5-I and VED in penile rehabilitation after laparoscopic NSRP for rectal cancer. Rectal cancer affects a younger population than PC; this allows us to study a younger population with optimal pre-intervention EF. The participants were divided into three groups: no treatment, overnight use of sildenafil 25 mg for 3 months after surgery, and concomitant use of sildenafil and VED 15 min/day for 3 months. The study reported a higher IIEF-5 score in the combination therapy groups. See Tables 1 and 2 for the studies characteristics.

Raina et al. is the only study that compared the efficacy of early VED with respect to no treatment after both nerve-sparing and non-nerve-sparing RP (NSRP). The results are that the use of VED after RP facilitates early sexual intercourse, early sexual satisfaction of the patient and spouse, and potentially an early return of natural erections sufficient for vaginal penetration. Results are superior in patients treated with the nerve-sparing technique [29], see Table 2.

Monga et al. evaluated the effect of early VED use on ED (measured by IIEF-5 score) and penile shortening after NSRP. Participants were divided into two groups: use of the VED beginning at 1 month and 6 months after the operation. The study suggests that the early onset of VED appears to be an effective strategy for improving ED and penile shrinkage. Also, when vacuum therapy is combined with PDE5-I, rehabilitation is more effective [30], see Table 2.

Unfortunately, the probability of regaining EF after RP has generally not improved in the last decade despite the improvement of surgical techniques and the introduction of new rehabilitation treatments. For this, it is necessary to standardize patient care after RP [38].

### 2.4. Intraurethral Alprostadil Treatment (IUA)

Intraurethral alprostadil (IUA) is a urethral treatment that delivers PGE1. PGE1 acts locally by increasing levels of cyclic adenosine $3',5'$-cyclinc monophosphate (cAMP), thus affecting the erectile tissue. Intraurethral alprostadil acts indirectly on the erectile tissue through the urethra. This urethral suppository bypasses the neural pathway in the corpora cavernosum and generally does not cause systemic side effects. The most common side effects reported are urethral burning and penile pain.

The first study for intraurethral alprostadil treatment used as a penile rehabilitation strategy was designed by Montorsi et al. in 1997. In this study, 30 patients who underwent bilateral NSRP were randomized to receive alprostadil injections 3 times per week for 12 weeks vs. no treatment. After 6 months, 67% of men in the treatment group vs. 20% in the control group achieved spontaneous erections. The study concluded that the injections of alprostadil decreased hypoxia-induced tissue damage. The therapy proved to be well-tolerated [39].

A lot of work in the past two decades focused on IUA treatment for erectile dysfunction after RP vs. other treatments. Recently (2017), Della Camera et al. [40] designed a prospective study that confirmed other previous prospective works that defined intraurethral alprostadil as a valid second line of treatment to common injective therapies in selected patients after RARP. Alprostadil was administered $\geq 2$, twice a week, and, at month 6, the IIEF-5 decreased from 20.5 preoperative to 18.1 post-treatment. The quality of life score decreased from an average of 5.1 to 2.3. Weekly sexual intercourse decreased from an average of 2.1 to 1.7 times. 89.7% of patients showed a positive SEP-Q2 and 77.8% showed a positive SEP-Q3. Of all 68 analyzed patients, 13 (17.6%) switched to intracavernous injection therapy.

Several studies have been designed to study combination therapy using a PDE5-I and alprostadil. In a recent review (2018), Moncada et al. [41] showed that combination therapy with a PDE5-I and alprostadil has been shown to be more effective than monotherapy

with a PDE5-I or alprostadil in patients who have previously failed either treatment as monotherapy. The potential risk of additive side effects does not appear to have an impact on patient compliance, while the additional benefit of synergistic smooth-muscle relaxant effect provides a useful supplement to the therapeutic armamentarium.

One of the most important facts about IUA is that it is an effective safety profile for treating patients with erectile dysfunction refractory to oral treatment with PDE5-I. The safety profile of PDE5-I is comprehensive and generated over two decades of use; current guidelines suggest them as the first-line treatment for ED. Patients who are candidates for intracavernous injections should be informed about the potential risk of priapism and penile fibrosis. Intraurethral and topical alprostadil is generally well-tolerated and patients should be informed about their benefits and risks/burdens. In general, the treatment regimen employed should be determined following a detailed examination of a patient's physical and psychosocial profile and tolerability while also fulfilling his wishes and expectations. The most common adverse effect was urethral burning. There were no cases of urinary tract infection, syncope, or priapism. These safety measures were recently collected in a review from Garrido Abad P et al. [42].

Very important data on this argument was collected by Philippou YA et al. in 2018. This study on several treatments for ED after robotic RP showed that scheduled, daily PDE5-I therapies may result in little or no effect on self-reported potency (RR 1.10, 95% CI 0.79, to 1.52; very-low quality evidence) vs. scheduled IUA at short-term follow-up. This work suggests that all single or combined treatments for post-robotic RP ED must be started earlier to have the max effect on erectile function [43].

## 2.5. Penile Vibratory Stimulation

The use of penile vibratory stimulation (PVS) was first described by Sobrero et al. in 1965 [44]. Advancements in technology and technique led to the development of devices that stimulate penile erection in men with ED and ejaculation in men with spinal cord injuries. PVS works through the stimulation of branches of the pudendal nerves along the penile shaft. The stimulation causes a reflex parasympathetic erection through the activation of nerve terminal endings that release nitric oxide and inhibit sympathetic fibers. The resultant effect is the liberation of cGMP and cAMP. Both of these cause cavernous smooth muscle dilation and penile engorgement [45].

The use of PVS was only to induce ejaculation in men with spinal cord injury for many years and, in 2014, Fode et al. pioneered the use of PVS as an agent in ED after nerve-sparing RP. This study had significant limitations, it showed that PVS is both acceptable and tolerable for patients, but not more [46].

In 2015 Fode and Sønksen tried to draw a randomized pilot study on PVS on post-robotic RP incontinence, but they described how PVS may have effects on EF and orgasms too. It was the first study that concluded that PSV could be included in post-RP treatments [47].

## 2.6. Low-Intensity Extracorporeal Shockwave Therapy (LIESWT)

In recent years, there has been an increasing interest in the use of LIESWT for ED. Hypoxia of the corpora cavernosa, secondary to ED, can result in a high expression of pro-fibrotic factors that, with subsequent development of corporal fibrosis, can cause ED. LIESWT technology exploits trauma and mechanical stress to stimulate the release of angiogenic factors, such as endothelial nitric oxide synthase (eNOS) and vascular endothelial growth factor (VEGF); these factors are responsible for tissue neovascularization in the penile tissue, improving penile blood flow by improving the erection [48–50].

The first to evaluate the effectiveness of LIESWT in improving ED without drug use was Vardi in 2010. In this first study, 20 patients with vasculogenetic ED were evaluated, at 6 months of follow-up, significant increases in IIEF scores were recorded in all men (20.9 ± 5.8 vs. 13.5 ± 4.1, $p < 0.001$) and significant increases were recorded in the duration of erections and penile rigidity, and significant improvement in penile endothelial

function were demonstrated. Ten men did not require any PDE5-I therapy after the 6 month follow-up. No pain was reported during the treatment and no adverse events were noted during the follow-up [51]. There are several LIESWT machines (electromagnetic, electrohydraulic, and piezoelectric) with different types of shockwave energy [52]. Several authors have studied the efficacy and safety of LIESWT in vasculogenetic ED but there is limited data on its efficacy in ED of patients treated for PC. The first to evaluate the efficacy of this treatment on PC therapy was Frey et al. in 2015. In the study, 16 patients, all having undergone bilateral robot-assisted NSRP, were treated with Duolith® SD1 T-Top (Storz Medical, Tägerwilen, Switzerland). This machine uses an electromagnetic system to generate shockwaves. The median baseline IIEF-5 score was 9.5 (range 5 to 20), with one patient using the "medicated urethral system for erections" (MUSE) and 11 using phosphodiesterase-5 inhibitors (PDE5i). At t1, the median change in IIEF-5 scores was +3.5. At t2, the median change from baseline IIEF-5 scores was +1. At t1 and t2, 11 and 7 patients, respectively, reported being either satisfied or very satisfied with the treatment. Only a few patients described mild pain when the shockwave got close to the urethra during LIESWT. In this, patients' recovery of EF was minimal [52,53]. In a randomized controlled trial study, 128 patients undergoing nerve-sparing radical cistoprostatectomy, with insufficient erection for vaginal penetration were available and categorized into three groups: PDE5-I, LIESWT, and a control group. Potency recovery rates at 9 months were 76.2%, 79,1%, and 60% in LIESWT, PDE5-I, and the control group, respectively. LIESWT is as safe as oral PDE5-I, but these differences are not statistically significant [54]. Baccaglini et al., in another randomized clinical trial, found an improvement of the IIEF5-5 score in patients undergoing LIESWT plus Tadalafil 5 mg/day vs. the control group undergoing only Tadalafil 5 mg/day; however, after RP, the difference was not clinically and statistically significant [55]. Contrary to this study, Inoue et al. showed that the sexual function score was significantly higher in patients undergoing early LIESWT compared to delayed or non-LIESWT groups [56].

*2.7. Stem Cell Therapy (SCT)*

In recent years, interest in stem cell therapy (SCT) in the rehabilitation of erectile dysfunction has been increasing, though the exact mechanism remains unclear [57]. In fact, in recent years there has been a growing number of scientific studies correlating the use of SCT in ED, even if SCT in post-treatment ED of prostate cancer is limited to a few studies. SCT are cells with different characteristics and functions; they are pluripotent, unique, capable of unlimited proliferation and perpetual self-renewal, proangiogenic antifibrotic, and antiapoptotic [57–59]. Existing different types of SC include the following: adipose-derived stem cells (ADSC), bone marrow-derived stem cells (BMSC), amniotic fluid stem cells (AFSC), urine-derived stem cells (USC), hematopoietic stem cells (HSC), adipose tissue stem cells (ADSC), and others [59,60]. Administration of SC is mainly done through intracavernous injection. BMSC was first reported as a source of mesenchymal stem cell (MSC) improved EF in rat models [61]. AFSC have also shown promise. Some studies show recovery of erection in mice with neurovascular pelvic injury after intra-cavernous injection of AFSC without serious side effects [62,63]. Yiou et al. performed the first non-randomized dose escalation pilot clinical trial that investigated the intra-cavernous injection of BMSC in patients with several post-RP ED refractory to medical therapy in two stages [63,64]. The primary objective of this phase was to assess the safety of four doses of intra-cavernous injection of BMSC used to treat post-RP ED ($2 \times 10^7$, $2 \times 10^8$, $1 \times 10^9$, and $2 \times 10^9$) in 12 consecutive patients within 6 months after the injection. The secondary objective was to evaluate sexual function, penile vascularization, endothelial function, and change in penile length. Results from stage I of this study showed no serious side effects, and sexual function and penile vascularization significantly improved 6 months after injection in 9 of 12 patients. Penile length was significantly increased in 1st and 3rd month but not at 6 months. Results from stage II showed similar results in erectile function and safety after intra-cavernous injection of BMSC. A recent clinical trial showed that the use of autologous

ADSC in a single dose in 21 patients improved erectile function in 53% of patients after 6 and 12 months [65,66].

## 3. Discussion

Prostate cancer is more commonly diagnosed in younger men than ever before. Today, PC treatments are various and provides excellent long-term oncologic outcomes [67]. An important problem for doctors remains the management of the negative sexual impacts this type of treatment has, such as erectile dysfunction, changes in the length of the penis, pain during intercourse, and lack of orgasm [4]. The relation between the enhanced recovery after RP and the benefit of a preoperative sexual rehabilitation program, also defined as pre-habilitation, has been studied in some different trials. The pre-habilitation should include information about the patients and the partners, with closer follow-up and the use of multimodal treatment approaches (oral medication, vacuum devices, pelvic floor muscle training, etc.), but clear clinical recommendations are not available yet [68].There are different types of therapy for penile rehabilitation post PC therapy. Surely the most used drugs are PDE5-I. Since 1997, when Montorsi introduced the concept of post-RP penile rehabilitation [39], a large number of works were done on the safety and functionality of PDE5-I. The REINVENT study, a double blind randomized trial involving 628 patients, showed that using vardenafil on-demand has better results on IIEF scores than daily use or placebo [12]. Also, avanafil use on-demand improved the EF of patients undergoing RP to a placebo [14]. Similar results were highlighted in the REACTT study. Patients undergoing NSRP benefited more from therapy with tadalafil 5 to 20 mg, daily or on-demand, respectively, compared to a placebo. This study also found that patients receiving tadalafil therapy had a better chance of preserving penile integrity, such as penile length, and maintaining a good quality of life compared to the placebo group [14–16]. Also, patients undergoing ADT or RT for treatment of PC benefit from PDE5-I therapy [19–21]. However, most of these studies showed that discontinuing therapy does not have any advantages compared to the placebo group. This review highlighted the efficacy of different PDE5-I uses after PC treatments, but the aspects that could play a very important role in the success of the therapy are when therapy is begun, the duration, the dosage, the drug used and the EF condition before treatment. In a recent review, Marchioni et al. highlighted that only a few studies focused on this aspect of treatment and the follow-up duration was heterogeneous [69]. An important tool for penile rehabilitation is also represented by the VED, in or without association with PDE5-I. Different studies showed that VED use increases the patient's confidence and sexual satisfaction and could be use as an alternative to patients who cannot take PDE5-I. VED has been shown to be useful after RP; however, there are no studies for patients treated with RT and/or ADT. PDE5-I and VED in combination are considered the first-line option for penile rehabilitation and ICI therapy is an important second line option [26]. ICI therapy has proven its worth in combination with PDE5-I and in patients who cannot take PDE5-I. The problem with this therapy is adverse events, such as local pain and priapism, which yield a high dropout rate, 11–31% [25]. Most recent studies on this topic are small and uncontrolled, which makes it difficult to make firm conclusions. Despite the range of physical and pharmaceutical interventions available, research on patients treated with RT and ADT has focused solely on the use of PDE5-I and there are no ICI-based rehabilitation studies [70]. In recent years, most studies and reviews of the literature on the administration of intraurethral alprostadil, even with some limitations, have shown to have positive effects on patients' erectile function post treatment of PC. Treatments with PDE5-I are still considered the first line of treatments in this field. Association with IUA are effective, safe, and tolerated by patients and must be considered earlier in post-surgery, but more studies are needed to strengthen this recommendation.

Technological development has led to the emergence of tools that allow for penile stimulation without the use of drugs. This is the case with PVS and LIESWT. To date, most of the studies use PVS in the stimulation of ejaculation in spinal cord injury and

only a few studies concern penile rehabilitation with PVS after RP. To our knowledge, no studies have been done on other types of PC treatment. More studies are needed for understanding the clinic value of these technologies in ED post-PC treatments. LIESWT were introduced by Vardi in 2010 to treat vasculogenetic ED without use of drugs. The easy use, the near absence of side effects, and the good results have allowed for the technique's rapid spread. However, many studies must be done to evaluate the safety and efficacy of LIESWT in vasculogenetic ED, but there are few contrasting studies that evaluate the efficacy of this technology in patients with ED post-PC treatment [51,56]. Even if the mechanism of action in not yet fully know, there is great interest in the use of SCT in ED. The first non-randomized clinical trial performed by Yiou in 2017 showed that this therapy is safe and that sexual function and penile vascularization improved 6 months after injection [63]. However, too few studies have been carried out to bring this treatment into clinical practice. All these rehabilitation techniques should be used in patients free of disease, with controlled PSA, in the absence of signs of disease recurrence.

Our study has some limitation. First, many studies contain bias and utilize a low number of patients. Moreover, many scientific works are based on the use of a specific therapy in RP and not in other types of treatment for PC. Finally, many studies lack a long-term follow-up and the eventual switch from one therapy to another. However, many studies, especially those regarding new technologies, will be necessary to find the best therapeutic scheme for penile rehabilitation, customized according to the characteristic of the patients and type of treatment for PC.

## 4. Conclusions

Today, the best therapy for erectile dysfunction post-PC treatment is PDE5i in association with or without VED, but the new therapies are promising. For this reason, further studies are needed.

**Author Contributions:** Conceptualization, R.C. and P.D.F.; methodology, R.C.; validation, R.C. and P.D.F.; researches and investigations D.C. and A.D.P.; resources, D.C. and A.D.P.; data curation, R.C., M.M.; writing—review, R.C., D.C. and A.D.P.; editing, S.F.; supervision and conflict resolution, L.S. and M.M. All authors have read and agreed to the published version of the manuscript.

**Funding:** This research received no external funding.

**Institutional Review Board Statement:** Not applicable.

**Informed Consent Statement:** Not applicable.

**Data Availability Statement:** No new data were created.

**Conflicts of Interest:** The authors declare no conflict of interest.

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
