# Peer review of "Penile Rehabilitation after Prostate Cancer Treatment: Which Is the Right Program?"

_2673-4397, doi:10.3390/uro3010009_

Round 1
Reviewer 1 Report
The paper is nice and well-organized. I have only minor comments about the methodology.
Please consider a flowchart (as a figure) about paper selection according to the PRISMA statement. Moreover, please specify which of the Authors performed the research and who resolves the conflict. Moreover, please insert the databases used for this review.
Reviewer 2 Report
1. Manuscript Overview: The manuscript is highly relevant to the field and provides a comprehensive overview of the past, current, and future directions of penile rehabilitation following prostate cancer treatment. The authors described the various technologies available, their benefits and drawbacks, and the need for improvement for the benefit of patients.
2. Abstract: There are a few minor grammatical and typographical errors in the abstract that should be fixed. Additionally, some sentences may need to be rewritten or re-paraphrased before publication.
3. Introduction, Methods, Discussion, and Conclusion: Each section is well-written, scientifically sound, succinct, and addressed the manuscript's objectives. The references are sufficient and include recent works. However, a few small grammatical corrections are required.
Reviewer 3 Report
The purpose of this review is to summarize the scientific evidence relating to the penile rehabilitation of patients undergoing prostate cancer therapy. Minor revision is required.
- when presenting Treatment of localized PC, please indicate that multimodal treatment are available.
- when presenting adverse events after surgery for PCa , please report urinary incontinence. for the scope please cite the following paper on its management (10.1038/s41391-022-00558-x)
- Chek typos
Reviewer 4 Report
The idea is very important! I believe that cancer survivors need an optimal, majestic and dignified life. I advice to start as soon as it is possible, even before cancer treatment as a part of prehabilitation, at least the psychological preparation. I advice to write few sentences about prehabilitation as well, and their limitation, too. see later.
Clarification is needed in the text. Brachytherapy is also a kind of radiotherapy, Authors meant external beam radiotherapy when they mentioned RT. In early case PC RT or ADT is required NOT RT and/or ADT!!!! Huge different.
Must also highlight, that these rehabilitation techniques should use just in verified cancer-free patients, controlled by PSA and imaging techniques, otherwise tumor cells can be spread by stimulation. Oral phosphodiesterase type 5 inhibitors may lead PSA elevation, so should carefully use, in my pracice we had serious problems, because we though that there was biochemical relapse, but skip the drug PSA normalized.
Congratulation for the nice work! Must speak about this very important but hidden problem.
